# Identification of Putative Quantitative Trait Loci for Improved Seed Oil Quality in Peanuts

**DOI:** 10.3390/genes15010075

**Published:** 2024-01-05

**Authors:** Pengju Hu, Jianan Zhang, Yahui Song, Xing Zhao, Xinxin Jin, Qiao Su, Yongqing Yang, Jin Wang

**Affiliations:** The Key Laboratory of Crop Genetics and Breeding of Hebei, Institute of Cereal and Oil Crops, Hebei Academy of Agricultural and Forestry Sciences, Shijiazhuang 050035, China; 15075957527@163.com (P.H.); nkyhsh@163.com (J.Z.); 13832366936@163.com (Y.S.); zhaoxing1202@163.com (X.Z.); jinxinxin1984@163.com (X.J.); 15733288090@163.com (Q.S.)

**Keywords:** peanut, oil quality, fatty acids, QTL, epistatic effects

## Abstract

Improving seed oil quality in peanut (*Arachis hypogaea*) has long been an aim of breeding programs worldwide. The genetic resources to achieve this goal are limited. We used an advanced recombinant inbred line (RIL) population derived from JH5 × KX01-6 to explore quantitative trait loci (QTL) affecting peanut oil quality and their additive effects, epistatic effects, and QTL × environment interactions. Gas chromatography (GC) analysis suggested seven fatty acids components were obviously detected in both parents and analyzed in a follow-up QTL analysis. The major components, palmitic acid (C16:0), oleic acid (C18:1), and linoleic acid (C18:2), exhibited considerable phenotypic variation and fit the two major gene and minor gene mixed-inheritance model. Seventeen QTL explained 2.57–38.72% of the phenotypic variation in these major components, with LOD values of 4.12–37.56 in six environments, and thirty-five QTL explained 0.94–32.21% of the phenotypic variation, with LOD values of 5.99–150.38 in multiple environments. Sixteen of these QTL were detected in both individual and multiple environments. Among these, *qFA_08_1* was a novel QTL with stable, valuable and major effect. Two other major-effect QTL, *qFA_09_2* and *qFA_19_3*, share the same physical position as *FAD2A* and *FAD2B*, respectively. Eleven stable epistatic QTL involving nine loci explained 1.30–34.97% of the phenotypic variation, with epistatic effects ranging from 0.09 to 6.13. These QTL could be valuable for breeding varieties with improved oil quality.

## 1. Introduction

Peanut (*Arachis hypogaea* L., 2n = 4x = 40) is primarily consumed as a source of vegetable oil in China due to the high oil content (40–56%) in its seeds. Peanut oil is composed of several major fatty acids: palmitic (C16:0), stearic (C18:0), oleic (C18:1), linoleic (C18:2), arachidic (C20:0), behenic (C22:0), and arachidonic (C20:1) acids. The ratio of saturated (C16:0, C18:0, C20:0, and C22:0) to unsaturated (C18:1, C18:2, and C20:1) fatty acids in peanut oil is generally around 1:4 [1]. However, a low percentage of saturated fatty acids is desirable for edible oils, as consumption of saturated fatty acids is associated with an increased risk of coronary disease [2], and diets with high levels of oleic acid can help reduce cholesterol and prevent arteriosclerosis and heart disease [3]. Therefore, the quality of peanut oil depends on its fatty acid composition, which affects its nutritional value, flavor, and stability. Since oleic acid and linoleic acid together comprise approximately 80% of the total fatty acids in peanut oil, the quality of peanut oil mainly depends on the oleic acid and linoleic acid contents of the seed [4]. Peanut oil can be divided into two categories, non-high oleic and high oleic, based on the proportion of oleic acid [5]. For example, non-high oleic peanut oil generally consists of 40–50% oleic acid and 30–40% linoleic acid, whereas high oleic peanut oil generally contains more than 75% oleic acid and less than 10% linoleic acid [6]. Breeding cultivars with high oil quality is a major objective of peanut breeding programs.

Fatty acid composition is a typical quantitative trait that is affected by poly-genes, the environment and genotype (G) × environment (E) interactions [7]; however, varieties with high fatty acid contents may be less affected by the environment. For instance, Singkham et al. (2010) examined the variation in fatty acid composition among peanut genotypes and the effects of G × E interactions on fatty acid composition in genotypes with high, intermediate and low oleic acid contents [8]. Genotypic variation was determined to be the main source of variation for fatty acid composition, and significant G × E interactions were identified in the low and intermediate oleic acid groups but not in the high oleic acid group. The formation of oil mainly occurs during the late stage of seed maturation. Oleic acid levels increase, while levels of other fatty acids decrease, during seed maturation [8]. Peanuts grow underground, making them more sensitive to the soil environment than aerial fruits. Compared with those grown under favorable conditions, the seeds of peanuts subjected to drought conditions at the end of the growing season show increased oleic acid contents but decreased linoleic acid contents that are significantly affected by G × E interactions [9]. By contrast, variation in soil temperature adversely affects the formation of both oleic and linoleic acids [10]. Hence, elucidating the genetic mechanisms underlying the fatty acid traits of peanut oil may help breeders develop elite cultivars.

Breeding cultivars with better oil quality through traditional breeding programs is expensive, inefficient, and time-consuming because measuring fatty acid contents can only be carried out after harvest. Therefore, there is an urgent need in modern breeding programs to determine how to rapidly diagnose traits of interest from germplasm resources and to develop new germplasm that are not influenced by complex environments. To achieve this breeding goal, much effort has focused on identifying valuable QTL and cloning functional genes to improve oil quality in several oil crops [11]. For example, more than 250 QTL for traits related to soybean (*Glycine max*) seed oil are documented in the public database SoyBase, https://www.soybase.org/ (accessed on 21 November 2023), and several hundred QTL have been reported in rapeseed (*Brassica napus*) [12], all of which have been mapped in many different populations and environments. Unfortunately, progress in peanuts has largely lagged behind other oil crops, such as soybean and rapeseed, with only a few studies having focused on mapping QTL associated with fatty acid composition [11,13,14,15,16]. For instance, 78 main-effect and 10 epistatic QTL were detected for oil content and oil quality traits using two recombinant inbred line (RIL) populations genotyped with simple sequence repeat (SSR) markers [11]. A total of 110 QTL for oil and protein content and fatty acid composition were mapped onto 18 peanut chromosomes, with the QTL *qA05.1* detected in four environments and showing major phenotypic effects on the contents of oil, protein, and six fatty acids [17].

In the current breeding program, two homozygous recessive genes controlling high oleic acid content, *FAD2A* and *FAD2B*, have been extensively applied and their genetic effect have been also well described in previous genetic studies [18]. However, the oleic quality also presented significant differences among cultivars harboring the genotypes *fad2A* and *fad2B*, respectively, that might be due to the existence of additional minor gene effects. Therefore, exploring additional QTL besides those associated with *FAD2A* and *FAD2B* should increase the efficiency of breeding new peanut cultivars with high oil quality. In this study, we aimed to expand the available peanut genetic resources to facilitate the breeding of new varieties with high oil quality. We used a previously constructed bi-parental RIL population [19] to (1) determine the main fatty acid components of peanut oil and their genetic basis and (2) map QTL with additive effects, epistatic effects, and QTL × environment interactions in multiple environments. Our results provide a basis for conducting marker-assisted selection (MAS) breeding to increase oil quality in peanuts.

## 2. Materials and Methods

### 2.1. Plant Materials and Field Conditions

Two parental peanut cultivars with obvious differences in oil quality, JiHua No.5 (JH5) and KaiXuan 01-6 (KX01-6), were used in this study. Both cultivars were bred by the Institute of Cereal and Oil Crops, Hebei Academy of Agricultural and Forestry Sciences, China. Construction of the RIL population used in this study was described previously [19]. To determine their oil quality traits, peanut seeds of each genotype of the RIL population, which consisted of 192 F_9_ lines, were individually harvested under natural field conditions in 2019–2021 in both Dishang (114.72° E, 37.94° N) and Nongchang_3502 (38.23° N, 134.39° E), China. Harvested pods were dried in the sun, and pods with shells were placed in cold storage (4 °C) to prevent degeneration. The trial environments were designated as follows: E1, Dishang 2021; E2, Nongchang_3502 2021; E3, Dishang 2020; E4, Nongchang_3502, 2020; E5, Dishang 2019; E6, Nongchang_3502 2019.

### 2.2. Sample Preparation and Fatty Acid Analysis

Dried seeds were ground into fine powder using an electronic grinder (KN-295 Knifetec, FOSS, Copenhagen, Denmark), passed through a 325 µm screen, and stored in plastic bags at −20 °C until analysis. Fatty acids were analyzed using gas chromatography (GC) as described previously, with minor modifications [20]. Briefly, oil was extracted from 100–150 mg peanut seed powder in approximately 1.9 mL n-hexane in a 2 mL centrifuge tube. The oil was trans-esterified to fatty acid methyl esters (FAMEs) using 500 μL 0.4 mol/L KOH–methanol solution. After 20 min of incubation in a water bath at 60 °C, the organic layer was separated from the aqueous layer. The organic layer containing methyl esters was transferred to an autosampler with moderate distilled water and n-hexane for GC analysis. FAME standard mix RM-3 (Sigma-Aldrich, St. Louis, MO, USA) was used to establish peak retention times. Each sample was run for 35 min. Seven fatty acids were measured in each sample: palmitic acid (C16:0), stearic acid (C18:0), arachidic acid (C20:0), gadoleic acid (C20:1), behenic acid (C22:0), oleic acid (C18:1), and linoleic acid (C18:2).

### 2.3. Statistical Analysis

Phenotypic variations among recombinant inbred lines as well as correlation of fatty acid traits were analyzed using the Performance Analytics package in R. The generalized heritability (*H*^2^) values of the seven fatty acids across six environments were estimated as described by Yang et al. [19] using the following formula: *H*^2^ = *σ_g_*^2^/(*σ_g_*^2^ + *σ_ge_*^2^/*n* + *σ_e_*^2^/*nr*), where *σ_g_*^2^ is the genetic variance component among the RILs, *σ_ge_*^2^ is the RIL × environment variance, *σ_e_*^2^ is the residual variance, *n* is the number of environments and *r* is the number of replicates. Student’s *t*-test was used to assess significant differences in the seven fatty acid traits using SPSS 19 software [21].

### 2.4. QTL Mapping Analysis

A genetic map was constructed in a previous study [19]. The constructed high-resolution genetic map covered 3706.382 cM, with an average length of 185.32 cM per linkage group, using 2840 polymorphic SNPs. The inclusive composite interval mapping (ICIM) method integrated in QTL IciMapping V4.1 software was used to calculate the phenotypic values of the seven fatty acids in six environments [19]. The presence of a putative QTL in a given genomic region was determined using an LOD score threshold of 2.5. Significant putative QTL were further checked via permutation tests with 1000 replications at a significance level of 0.05. The parameters were set as follows: to detect additive QTL, two threshold methods were used. The step size was 1.0 cM, PIN was 0.001 and the Type I error was 0.05. To detect epistatic QTL, the permutation test method was used with 1000 permutations. The step size was 5.0 cM, the PIN was 0.0001 and the Type I error was 0.05. The finalized QTL profiles were depicted using MapChart 2.2 [22].

## 3. Results

### 3.1. Detection and Variation of Major Fatty Acids in the Seeds of the Two Parental Genotypes

Using GC, we could clearly detect and distinguish seven fatty acid components in the seeds of both parental genotypes (JH5 and KX01-6), including the saturated fatty acids palmitic acid (C16:0), stearic acid (C18:0), arachidic acid (C20:0), and behenic acid (C22:0), as well as the unsaturated fatty acids oleic acid (18:1), linoleic acid (18:2) and peanut monoenoic acid (C20:1) (Figure 1A,B). However, seeds from the two field-grown parents showed contrasting oil quality (Figure 1). A mixture of C16:0 (11.61%), C18:1 (42.02%) and C18:2 (35.72%) comprised the major proportion of fatty acids in JH5 seeds, whereas C18:1 alone comprised more than 80.87% of the fatty acids in KX01-6 seeds. Furthermore, oil from JH5 contained significantly higher (*p* < 0.05) C16:0 (Figure 1C), C18:0 (Figure 1D), C18:2 (Figure 1F) and C20:0 (Figure 1G) contents than that from KX01-6, with proportions 50.73%, 51.18%, 90.68% and 61.21% higher in JH5 than in KX01-6, respectively. By contrast, JH5 oil contained significantly lower (*p* < 0.05) C18:1 (Figure 1E) and C20:1 (Figure 1H) contents than KX01-6 oil, with 48.04% and 60.08% lower proportions in JH5 than in KX01-6, respectively. Proportions of C22:0 were similar in both parental genotypes (Figure 1I). These results indicate that oil quality differs significantly between JH5 and KX01-6 seeds.

### 3.2. Phenotypic Variations among Recombinant Inbred Lines

To facilitate QTL analysis, we evaluated phenotypic variations in these fatty acid traits among individuals of an available RIL population under two field conditions from 2019 to 2021 [19]; the results are summarized in Table 1. We observed significant phenotypic variation and extensive transgressive heritability for each of the fatty acid traits within this RIL population. The mean population value for each trait fell between the mean values of the parents, while the maximum and minimum values fell beyond the extremes of the parental values. These results indicate that genetic variation exists between the two parents, which is required for QTL identification. Furthermore, the values of all seven tested traits calculated over six environments fit normal distributions, as their absolute kurtosis and skewness values were <2.1. Broad-sense heritability (*h*^2^*_b_*) for the fatty acid traits observed over six environments varied from 0.876 to 0.987 (Table 1), indicating that the phenotypic variation observed among RILs in this population was mainly derived from genetic variation. The coefficient of variation (CV%) of four fatty acid components (C16:0, C18:1, C18:2 and C20:1) exceeded 24%, especially that of C18:2 (58.21%), and the CV% values of the three other fatty acid components (C18:0, C20:0 and C22:0) exceeded 14%. These data strongly suggest that extensive genetic variation underlies the fatty acid traits in this population. Significant phenotypic variations derived from genetic variation of the RILs were observed over six environments, which is required for the identification of QTL for these traits in peanuts.

### 3.3. Correlation Analysis of Fatty Acid Traits

The values for all tested traits were significantly correlated with each other (Figure 2): the absolute correlation coefficients ranged from 0.10 to 0.99 (*p* < 0.01). Interestingly, any two of the components C16:0, C18:1 and C18:2 had extremely high correlation coefficients, with values ranging from 0.97 to 0.99, suggesting they might be regulated by similar genetic components. Moreover, the frequency distributions of C16:0, C18:1 and C18:2 exhibited a bimodal pattern (Figure 2); there were 55 and 137 RIL families per peak. In χ2 tests, there was no significant difference (*p* value = 0.243 > 0.05) between these numbers and the expected 1:3 or 3:1 phenotypic segregation ratios of traits controlled by two major genetic loci. These results, together with the results of skew and kurtosis analyses, indicate that the genetic basis of C16:0, C18:1 and C18:2 fits the major gene and minor gene mixed inheritance model. Interestingly, the correlation coefficients showed significant differences between the families of the two peaks. We therefore divided the RILs into two groups, Peak 1 (n = 55) and Peak 2 (137), based on the bimodal features of C16:0, C18:1 and C18:2, for further correlation analysis (Table 2). For the Peak 1 group, C16:0 and C18:1 exhibited significant correlations with four and three other fatty acids, respectively, with correlation coefficients ranging from 0.16 to 0.58, whereas no significant correlation was observed between C18:2 and the four other fatty acids. By contrast, for the Peak 2 group, relatively weak correlations were observed for C16:0 and C18:1 with four other fatty acids, with correlation coefficients ranging from 0.07 to 0.24. However, C18:2 exhibited significant correlations with three fatty acids, with correlation coefficients of 0.21, 0.10 and 0.18, respectively. The obvious differences in correlation coefficients between the two groups strongly suggest that gene-to-gene interactions control fatty acid formation in peanut oil.

### 3.4. QTL Analysis of Fatty Acids in Individual Environments

We employed the ICIM-ADD method integrated in QTL ICiMapping software for QTL analysis in each environment. We identified 17 QTL for the seven fatty acid traits (Table 3; Figure 3), which explained 2.57–38.72% of the phenotypic variation observed among the RILs grown in six environments. Three QTL (i.e., *qFA_08_1*, *qFA_09_2*, and *qFA_19_3*) were simultaneously detected in all six environments. *qFA_08_1* in linkage group 8 accounted for phenotypic variations in three fatty acid traits, with percentages of phenotypic variation explained ranging from 9.08 to 22.43 and with LOD values between 6.33 and 15.58. Additionally, the two most stable QTL, *qFA_09_2* and *qFA_19_3* in linkage groups 9 and 19, respectively, could explain phenotypic variations for almost all major fatty acid traits, with percentages of variation explained ranging from 6.37% to 38.72% and with LOD values between 5.12 and 37.56. These results suggest that *qFA_08_1*, *qFA_09_2* and *qFA_19_3* are stable, valuable, major-effect QTL that could be applied in peanut breed programs to improve oil quality. Two QTL distributed in linkage groups 5 and 12, respectively, could be simultaneously detected in three environments and explained more than 10% of the phenotypic variation in at least one environment. For example, *qFA_05*, which is located in linkage group 5 and was detected in environments E1, E3, and E4, could explain 11.63–12.40% of the phenotypic variation for C20:0, with LOD values between 6.13 and 31.03. *qFA_12*, a QTL with LOD values of 5.59–9.80 that explained 6.11–17.13% of phenotypic variation for C20:0 and C22:0 in environments E1, E5 and E6, was mapped to linkage group 12. Besides, the other highest QTL (12) had minor effects, as they could explain 2.57–11.80% of variation for only one or two fatty acid traits in fewer than three environments.

### 3.5. Joint Analysis of Fatty Acid Traits in Multiple Environments

To further explore QTL that depend on the environment, we employed the MET module in QTL IciMapping V4.1 software for joint analysis based on observation of the RIL population grown in six environments. Thirty-five significant QTL associated with seven fatty acids were identified with LOD values ranging from 5.99 to 150.38 (Appendix A and Figure 3), which explained 0.94–32.21% of the phenotypic variation. All the QTL except *qFA_10* detected from individual environments were also found via joint analysis (Figure 4). However, the joint mapping method detected more QTL than those identified in individual environments. The contribution of additive QTL effects in the joint analysis was lower than that in individual environments. In addition, nine QTL, with LOD values ranging from 6.40 to 42.81, exhibited strong interactions with the environment. The contribution rates of additive (A) effects and additive × environment (AE) effects ranged from 1.04% to 10.99% and 1.07% to 20.52%, respectively, (Table 4). Interestingly, these environment-dependent QTL mainly explained the phenotypic variation of C18:0, C20:0, C20:1 and C22:0, which account for a relatively small proportion of fatty acids present in the seed. However, *qFA_05* had a high AE effect (20.52%) and is considered to be a major QTL for improving C22:0; its favored environment was E3.

### 3.6. Detection of Stable Epistatic QTL in Individual-Environment Trials

To further explore the epistatic effects of QTL on the quality of peanut oil, we analyzed the epistatic QTL with the ICIM-EPI mapping method using the BIP module in ICiMapping V4.1. This analysis returned 11 stable epistatic QTL that could be detected in at least two environments (Table 5). A total of nine loci were involved, among which four interacted with only one locus, whereas each of the other loci interacted with more than two loci. These nine loci involved in QTL interactions were distributed on chromosomes 8, 9 (2), 10, 12, 13, 14, and 19 (2), respectively. Each of these 11 stable epistatic QTL could explain 1.30–34.97% of phenotype variation in the fatty acid traits, with epistatic effects (additive by additive) ranging from 0.09 to 6.13. Furthermore, *qFA_E9_3*, and *qFA_E1*9, which were simultaneously detected in five environments, were considered to be major QTL with epistatic effects, which explained 8.02–34.97% and 4.98–25.01% of phenotype variation for four fatty acid traits (C16:0, C18:1, C18:2, and C20:1), respectively. These results implied that simultaneously selecting multiple genes maybe more helpful in further improving the quality of peanut oil.

## 4. Discussion

High-oleic peanut oil is generally considered to be beneficial to human health due to its association with lower blood serum cholesterol, especially low-density lipoprotein (LDL) [23]. In addition, high-oleic oil and high-oleic peanut products have a more desirable flavor and longer shelf life due to a slower decline in roasted flavor during storage and less off-flavor development than peanuts with typical oleic acid contents [24]. Therefore, breeding new peanut cultivars with high oleic acid contents has become a major breeding objective worldwide. Kaixuan 01-6 (KX01-6) is the mainstay parent for breeding peanut cultivars with high oleic acid content in China. Almost all peanut varieties currently grown in China with high oleic acid contents were derived from KX01-6 or its sister lines. To date, no studies on the genetic basis or molecular mechanism underlying the high oleic content of KX01-6 have been described. Therefore, in this study, we tried to identify the major loci affecting high oleic acid content in KX01-6 as well as minor-effect loci affecting oil quality to facilitate further MAS breeding.

Genetic analysis of the seven observed traits in our mapping RIL population revealed that three fatty acids (C16:0, C18:1, and 18:2) exhibited considerable phenotypic variation and fit the two major gene and minor gene mixed inheritance model. Importantly, two recessive mutant genes (*fad2A* and *fad2B*) are well known to have major effects on oleic acid and linoleic acid contents in peanuts [25,26,27]. To date, only these two genes have been shown to significantly increase the C18:1 content (from ~40% to ~80%) [28]. We therefore hypothesize that KX01-6 contains *fad2A* and *fad2B* or their recessive alleles, which underlies the considerable phenotypic variation in this trait. We observed significant correlations among all seven fatty acid traits (Figure 2). However, this might be expected since these fatty acids are metabolized in the same fatty acid metabolism pathway in plants. By contrast, the correlation coefficients among the seven observed traits differed considerably between the Peak1 and Peak2 groups (Table 2), perhaps due to the different genotypes of *FAD2A* and *FAD2B*. These results suggest that the metabolism of one fatty acid component may be affected by or dependent on the other fatty acid components; that is, a complex epistatic effect likely exists.

To uncover as many stable QTL behind fatty acid traits as possible, which will facilitate oil quality improvement, we used an RIL population and a high-resolution genetic map to identify QTL with additive effects, epistatic effects, and QTL × environment interactions across multiple environments. We identified two major QTL, *qFA_09_2* and *qFA_19_3*, on chromosomes 9 and 19 with physical positions consistent with those of *FAD2A* and *FAD2B* [19], respectively. We also sequenced haplotypes of *FAD2A* and *FAD2B* in KX01-6 [29]. Not surprisingly, both are loss-of-function haplotypes sharing 100% sequence identity with *fad2A* and *fad2B*, respectively. These results suggest that existing peanut varieties with high oleic acid content are all dependent on the same genetic resources, at least in China. However, due to linkage disequilibrium, the genetic diversity around *fad2A* and *fad2B* might be very low, making it difficult to improve *fad2A-* or *fad2B*-linked traits [19]. We also identified another stable QTL, *qFA_08_1*, with relatively large effects (PVE% = 9.08–22.43%) on chromosome 8. This QTL, unlike the two other stable QTL, only influences three fatty acid traits (C18:0, C20:0 and C20:1) and maps close to our previously reported QTL *qOC08_2* by using the same RIL population [19], which significantly increases seed oil content. Taken together, these results suggest that *qFA_08_1* affects the quality of peanut oil by increasing or decreasing the absolute contents of some fatty acids in seeds, whereas *qFA_09_2* and *qFA_19_3* mainly increase the quality of peanut oil by affecting components of the biosynthetic pathway in fatty acid metabolism.

Peanut is an allopolyploid crop that likely originated from hybridization between two wild diploid species, *Arachis duranensis* and *Arachis ipaensis*, followed by chromosome doubling [30]. The relatively recent divergence of *A. duranensis* and *A. ipaensis* approximately 2.5–3.5 million years ago resulted in a strong similarity between the sub-genomes of cultivated peanut [25,31], which carry a large number of duplicated genes with similar functions. *FAD2A* and *FAD2B*, both of which encode fatty acid desaturase, originated from *A. duranensis* and *A. ipaensis*, respectively, and share 99.20% sequence identity. Therefore, it is not surprising that these genes completely or partially complement each other’s functions. For example, in this study, we divided the RILs into four groups based on the presence of *FAD2A* and *FAD2B* alleles to examine variation in observed oleic acid content. When the two mutant alleles were present together, the oleic acid content was approximately 30% higher than that in genotypes possessing only one of these alleles (Appendix A). By contrast, the oleic acid content did not significantly differ between the two groups harboring the genotypes *FAD2A/fad2B* and *fad2A/FAD2B*, respectively. We also detected an additive effect between *FAD2A* and *FAD2B*, as indicated by the finding that the oleic acid content was approximately 10% lower in genotypes harboring both *FAD2A* and *FAD2B* than in genotypes possessing only one of these alleles (Appendix A). However, the additive effect was not due to the simple addition of *FAD2A* and *FAD2B*, pointing to an epistatic effect between *FAD2A* and *FAD2B*. Indeed, our epistatic QTL analysis confirmed this hypothesis (Table 5).

In summary, the fatty acid metabolism pathway in peanuts is a complex process that involves many genetic loci. Although *FAD2A* and *FAD2B* are employed in breeding high-oleic-acid peanut cultivars, many QTL with epistatic effects or environment interactions have not attracted breeders’ attention. Therefore, the precise breeding of peanut cultivars with high oil quality has not yet been achieved. Our results provide additional genetic resources that can be utilized for the precise breeding of this trait in the future.

## 5. Conclusions

In this study, we analyzed seven main fatty acids components in an RIL population. The major components, palmitic acid (C16:0), oleic acid (C18:1) and linoleic acid (C18:2), exhibited considerable phenotypic variation and fit the two major gene and minor gene mixed inheritance model. We further identified seventeen QTL from six individual environments, thirty-five QTL for multiple environments, and sixteen QTL were detected from both individual and multiple environments, including a novel QTL *qFA_08_1* with stable, valuable and major effect. In addition, we also found eleven stable epistatic QTL involving nine loci, and the epistatic QTL between *qFA_E9_3* and *qFA_E19* explained the main phenotype variation for four fatty acid traits (C16:0, C18:1, C18:2, and C20:1). Overall, our study provided some valuable information for breeding varieties with improved oil quality.

## Figures and Tables

**Figure 1 genes-15-00075-f001:**
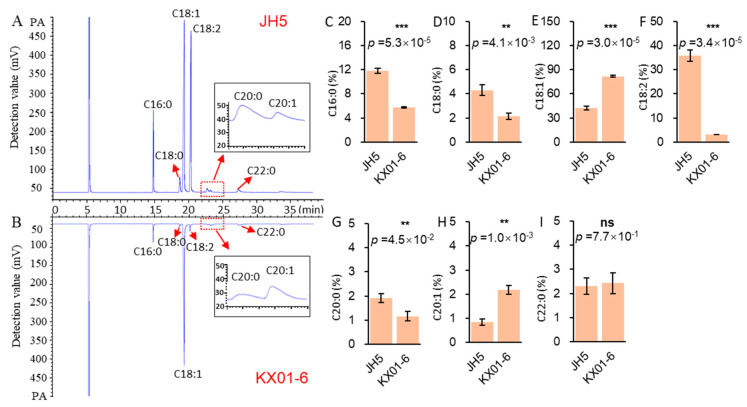
Analysis of main fatty acid components in JH5 and KX01-6. (**A**,**B**) were UPLC chromatograms of JH5 and KX01-6, respectively. C16:0—palmitic; C18:0—stearic; C18:1—oleic; C18:2—linoleic; C20:0—arachidic; C20:1—arachidonic; C22:0—behenic. (**C**–**I**) were different analysis for C16:0, C18:0, C18:1, C18:2, C20:0, C20:1, and C22:0, respectively, between parent. Asterisks indicate the significance of differences between JH6 and KX01-6 in Student’s *t*-test at 1% (**) and 0.1% (***) level, and “ns” represents no significant differences.

**Figure 2 genes-15-00075-f002:**
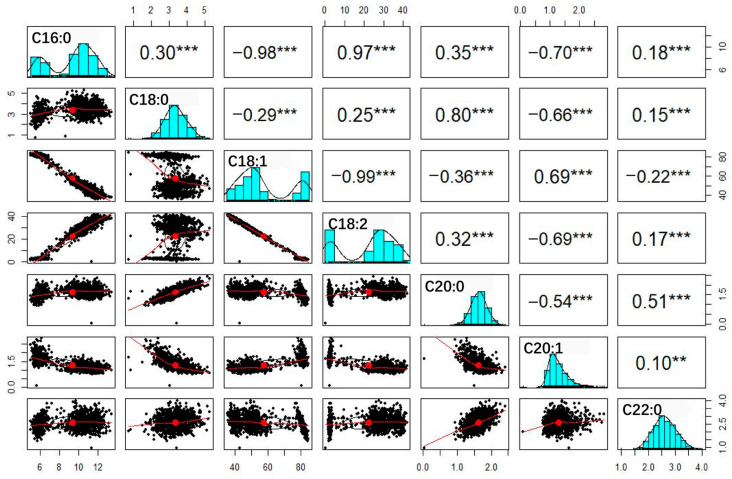
Correlation analysis among the 7 observed traits. The histograms with fitting curve of traits are shown diagonally. Above the diagonal is correlation coefficient with probability level; below the diagonal is scatter plot with fitting curve. Asterisks indicate the significance of differences in RILs through Student’s *t*-test at 1% (**) and 0.1% (***) probability level.

**Figure 3 genes-15-00075-f003:**
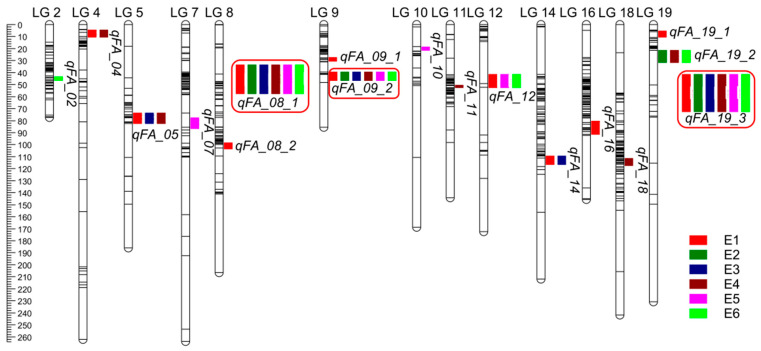
QTL associated with fatty acid traits detected in RIL population from six individual environments. The red box means the QTL were simultaneously detected in all six environments.

**Figure 4 genes-15-00075-f004:**
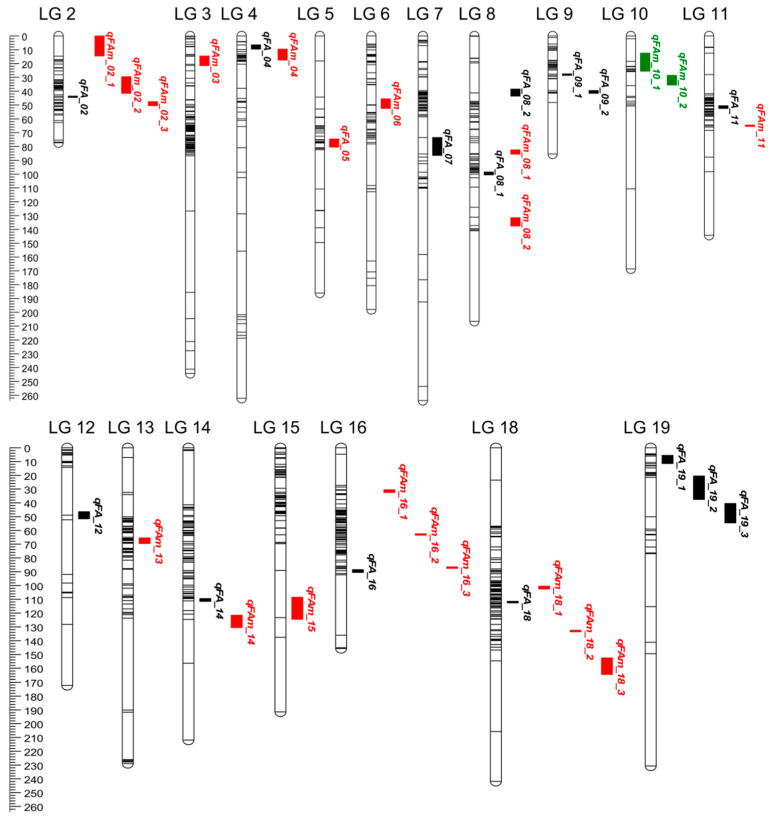
Additive QTL × the environment interaction effects for the fatty acid traits in RIL population across six environments. The QTL with red, green and black color mean QTL could be detected in multiple, individual and both environments, respectively.

**Table 1 genes-15-00075-t001:** Description of phenotype analysis for seven fatty acid components in the RIL population across six environments.

Traits	Parents		RILs							*H* ^2^
	JH5	KX01-6	Mean	SD	CV%	Max	Min	Kurt	Skew	
C16:0	11.61	5.72	9.32	2.27	24.32	13.40	4.94	−1.09	−0.55	0.986
C18:0	4.24	2.07	3.36	0.60	17.82	5.29	0.75	0.35	0.01	0.932
C18:1	42.02	80.87	57.71	15.36	26.62	84.78	35.51	−1.13	0.61	0.987
C18:2	35.72	3.33	22.57	13.14	58.21	41.97	0.41	−1.16	−0.58	0.987
C20:0	1.87	1.16	1.63	0.23	14.26	2.42	0.04	1.75	−0.26	0.921
C20:1	0.91	2.28	1.29	0.33	25.69	2.82	0.08	2.06	1.25	0.966
C22:0	2.42	2.66	2.61	0.43	16.34	4.01	0.99	0.00	0.17	0.876

Note: SD—standard deviation; CV%—coefficient of variation; Max—maximum value; Min—minimum value; Kurt—kurtosis; Skew—skewness and *H*^2^—broad-sense heritability. The mean values derived from the RILs population consisted of 192 family lines.

**Table 2 genes-15-00075-t002:** Correlation analysis of observed traits in the different distribution peak.

Fatty Acid	Group	C18:0	C20:0	C20:1	C22:0
C16:0	Peak1	0.34 ***	0.31 ***	−0.16 **	0.28 ***
	Peak2	−0.07 *	−0.05 ^ns^	−0.24 ***	0.00 ^ns^
C18:1	Peak1	−0.31 ***	0.47 ***	−0.08 ^ns^	−0.58 ***
	Peak2	0.12 ***	0.03 ^ns^	0.17 ***	−0.09 **
C18:2	Peak1	−0.01 ^ns^	0.01 ^ns^	0.03 ^ns^	0.07 ^ns^
	Peak2	−0.21 ***	−0.10 **	−0.18 ***	0.01 ^ns^

Note: ^ns^ Non-significant. * Significance at the 0.05 probability level. ** Significance at the 0.01 probability level. *** Significance at the 0.001 probability level.

**Table 3 genes-15-00075-t003:** QTL associated with the seven fatty acid trait detected in individual environments.

QTL	Fatty Acid	LG	Environment	Position	Left Marker	Right Marker	LOD	PVE%	Add	Threshold
*qFA_02*	C20:0	2	E6	44	Chr02.92004332	Chr02.92097568	4.63	4.28	0.05	4.20
*qFA_04*	C18:0, C20:0	4	E1, E4	7–8	Chr04.7737038	Chr04.10252892	4.48–5.90	5.60–6.90	0.05–0.15	4.20–4.40
*qFA_05*	C20:0	5	E1, E3, E4	75–82	Chr05.37478237	Chr05.88563212	6.13–31.03	11.63–12.40	0.07–0.33	4.50–8.00
*qFA_07*	C22:0	7	E5	83	Chr07.4150878	Chr07.3251821	4.71	8.50	0.10	4.30
*qFA_08_1*	C18:0, C20:0, C20:1	8	All	39–55	Chr08.42575311	Chr08.46776535	6.33–15.58	9.08–22.43	0.10–0.29	4.20–6.60
*qFA_08_2*	C22:0	8	E1	100	Chr08.15237857	Chr08.14658317	5.48	6.14	0.08	4.80
*qFA_09_1*	C18:0	9	E1	28	Chr09.111222696	Chr09.111444555	9.35	11.57	0.17	4.40
*qFA_09_2*	C16:0, C18:0, C18:1, C18:2, C20:0, C20:1	9	All	40–41	Chr09.113696781	Chr09.114431570	5.12–32.93	6.37–28.79	0.05–9.65	4.00–6.60
*qFA_10*	C18:2	10	E5	19	Chr10.5820536	Chr10.6895328	4.23	3.21	2.72	4.20
*qFA_11*	C22:0	11	E4	51	Chr11.135590338	Chr11.134662327	5.35	9.89	0.09	4.50
*qFA_12*	C20:0, C22:0	12	E1, E5, E6	49–50	Chr12.17574379	Chr12.3375707	5.59–9.80	6.11–17.13	0.04–0.11	4.30–4.70
*qFA_14*	C20:0	14	E1, E3	110	Chr14.139524574	Chr14.137319755	5.14–18.38	5.52–11.80	0.04–0.16	4.30–13.30
*qFA_16*	C20:0	16	E1	89	Chr16.147123562	Chr16.146960299	6.18	6.69	0.05	4.30
*qFA_18*	C18:0	18	E4	112	Chr18.114667904	Chr18.115066398	5.58	6.97	0.16	4.20
*qFA_19_1*	C18:2	19	E1	7	Chr19.8567191	Chr19.11838675	4.12	2.57	2.51	4.10
*qFA_19_2*	C16:0, C18:1, C18:2	19	E2, E4, E6	11–20	Chr19.12239093	Chr19.12852116	4.02–10.14	2.64–7.50	0.47–5.27	4.00–4.10
*qFA_19_3*	C16:0, C18:1, C18:2, C20:0, C20:1	19	All	39–46	Chr19.138748208	Chr19.156112831	5.99–37.56	9.14–38.72	0.07–11.35	4.00–6.60

Note: LG—linkage group; LOD—logarithm of odds; PVE—phenotypic variation explained by QTL; Add—the additive effect. The same below. The threshold values were calculated using the 1000 times permutation test.

**Table 4 genes-15-00075-t004:** Additive QTL × the environment interaction effects for the fatty acid traits in RIL population.

QTL	LG	Trait	Position	Marker Interval	LOD	PVE%		Add	Interaction Effect between Additive QTL and Environment					
						A%	AE%		AE1	AE2	AE3	AE4	AE5	AE6
*qFA_02*	2	C20:0	44	Chr02.92004332_Chr02.92097568	6.40	1.04	1.07	1.50	−0.06	−0.06	−1.74	−1.41	0.30	2.96
*qFAm_04*	4	C20:1	17	Chr04.87381676_Chr04.80904224	38.42	4.14	4.67	−7.16	7.26	−15.79	3.91	−1.36	5.20	0.79
*qFA_05*	5	C22:0	75	Chr05.90854275_Chr05.88563212	42.81	10.99	20.52	8.53	2.66	−4.53	24.71	−6.35	−8.44	−8.05
	5	C20:0	78	Chr05.41888224_Chr05.37841097	11.34	2.16	2.76	2.16	−2.07	0.62	−2.23	4.16	−2.32	1.83
*qFA_08_1*	8	C18:0	42	Chr08.46776535_Chr08.44595184	16.76	3.94	4.57	7.75	−5.56	−2.39	−5.57	18.37	−2.14	−2.71
*qFA_09_1*	9	C18:0	28	Chr09.111222696_Chr09.111444555	11.26	1.86	2.11	5.32	12.09	−5.57	−1.90	−3.29	−0.69	−0.63
*qFA_12*	12	C22:0	49	Chr12.17574379_Chr12.3375707	19.29	2.43	4.86	−4.01	−6.93	3.96	4.06	3.99	−9.02	3.94
*qFAm_16_3*	16	C22:0	87	Chr16.147687093_Chr16.147523996	6.70	1.08	1.25	−2.94	−5.20	−3.32	3.82	1.66	1.83	1.20
*qFA_16*	16	C20:0	89	Chr16.147123562_Chr16.146960299	8.93	1.07	1.59	−1.62	−2.84	−1.48	3.50	0.01	−0.09	0.90

Note: AE1–AE6, interaction effects between additive QTL and individual environments. The QTL with “FAm” were only detected in multiple environments; the others were detected in both individual and multiple environments. A%, phenotype variation caused by the additive effect; AE% phenotype variation caused by interaction effects between additive QTL and individual environments.

**Table 5 genes-15-00075-t005:** Stable epistatic QTL for fatty acid in the RIL population.

Epistatic QTL	Environment	Trait	Loc.1	Pos.1	Marker interval	Loc.2	Pos.2	Marker interval	LOD	PVE%	Add1	Add2	Add by Add
*qFA_E8*	E1, E3	C16:0, C18:1	8	170	Chr08.2179124_Chr08.43274117	9	40	Chr09.113696781_ Chr09.115494682	5.00–6.66	4.51–7.32	0.67–5.02	1.22–8.84	0.67–3.96
*qFA_E9_1*	E1, E3	C16:0, C18:1, C18:2	9	40	Chr09.113696781_ Chr09.115494682	9	70	Chr09.115494715_Chr09.112209867	5.24–9.30	4.65–13.79	1.20–8.93	0.51–3.74	0.84–6.13
*qFA_E9_2*	E1, E3	C16:0, C18:1, C18:2	9	40	Chr09.113696781_ Chr09.115494682	10	80	Chr10.20573531_Chr10.2981198	5.01–8.87	4.55–13.99	1.21–8.82	0.54–3.93	0.82–5.37
*qFA_E9_3*	E1, E2, E3, E4, E5	C16:0, C18:1, C18:2, C20:1	9	40	Chr09.113696781_ Chr09.115494682	19	35	Chr19.138748208_Chr19.156112831	5.92–28.01	8.02–34.97	0.00–9.57	0.13–10.95	0.09–4.30
*qFA_E9_4*	E2, E4, E5, E6	C16:0, C18:1, C18:2	9	45	Chr09.113696781_Chr09.115494682	9	70	Chr09.115494715_Chr09.112209867	5.10–8.43	5.21–9.02	0.10–0.83	0.50–4.17	0.87–5.85
*qFA_E9_5*	E4, E6	C16:0, C18:1, C18:2	9	45	Chr09.113696781_Chr09.115494682	10	80	Chr10.20573531_Chr10.2981198	5.29–6.41	5.94–7.64	0.00–0.14	0.70–4.84	0.76–5.63
*qFA_E9_6*	E5, E6	C16:0, C18:1, C18:2	9	45	Chr09.113696781_Chr09.115494682	12	145	Chr12.3561394_Chr12.116523714	5.05–5.74	4.54–6.82	0.22–0.87	0.62–4.04	0.75–5.08
*qFA_E9_7*	E4, E5, E6	C16:0, C18:1, C18:2	9	45	Chr09.113696781_Chr09.115494682	13	155	Chr13.143629587_Chr13.138922790	5.15–5.89	4.91–6.64	0.04–0.33	0.71–5.05	0.73–4.61
*qFA_E10*	E1, E3	C16:0, C18:1, C18:2	10	80	Chr10.20573531_Chr10.2981198	19	145	Chr19.155910347_Chr19.153836481	5.29–7.25	1.40–4.10	0.56–4.04	0.12–0.89	0.78–5.21
*qFA_E14*	E1, E4	C16:0, C18:1, C20:1	14	190	Chr14.13558383_Chr14.5680674	19	145	Chr19.155910347_Chr19.153836481	5.05–6.64	1.30–9.28	0.06–4.23	0.04–0.36	0.13–4.36
*qFA_E19*	E1, E3, E4, E5, E6	C16:0, C18:1, C18:2, C20:1	19	40	Chr19.138748208_Chr19.156112831	19	145	Chr19.155910347_Chr19.153836481	6.54–29.13	4.98–25.01	1.49–11.07	0.00–0.83	0.09–4.81

Note: Loc.—Locus; Pos.—Genetic Position; Stable epistatic QTL means this QTL could be detected under two or more environments. Add by Add, epistatic effect of additive × additive.

## Data Availability

All of the original re-sequencing data used in this study have been submitted to the public database of GSA (Genome Sequence Archive) with GSA accessions numbers CRA007578 and CRA007525, respectively.

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
