# Peer review of "Identification of Putative Quantitative Trait Loci for Improved Seed Oil Quality in Peanuts"

_genes, 2024, doi:10.3390/genes15010075_

Round 1
Reviewer 1 Report
Comments and Suggestions for Authors
The manuscript outlines a study focused on identifying QTL related to seed oil quality in peanuts. The methodology is appropriate, and the results are effectively presented. However, the novelty of the study needs to be better articulated, and it remains unclear whether any new QTLs were identified. I recommend accepting the manuscript with minor revisions. Please refer to the attached file for my detailed comments and suggestions.

The quality of English language is generally good but requres moderate revision, especially some of the paragraphs needs to be reconstructed.
Reviewer 2 Report
Comments and Suggestions for Authors
The article entitled "Identification of QTLs for improved seed oil quality in peanuts" is reliable and provides important information on the genetic basis of the high content of oleic fatty acid in peanuts. Nevertheless, it has not only strengths, but also weaknesses that need to be strengthened.
Review:
1. Abstract Structure: The abstract is well-structured.
2. Introduction: The introduction provides a good overview of the research topic. However, the aim of the study should be better explained, or the authors should propose an alternative research hypothesis in relation to the null hypothesis and verify it in the later part of the work.
3. Materials and Methods: The chapter is generally well-presented but needs better structuring.
4. Discussion: The discussion requires more in-depth analysis and should be subdivided for better understanding.
5. Conclusions: Conclusions should be generalizing and summarizing.
Strengths:
1. Clear Statistical Analysis: The text includes a clear statistical analysis related to the fatty acid traits in peanuts. The introduction of information such as correlation coefficient values and QTL analysis in different environments provides robust scientific data.
2. Precise Presentation of Results: Research results are accurately presented in the form of tables and graphs, facilitating reader understanding. Key information, such as QTL locations on chromosomes, percentages of explained phenotypic variability, and additive effects, is transparently presented.
3. Usefulness for Breeders: Studies on the genetic basis of high oleic acid content in peanuts are valuable for plant breeders. Indicated genes influencing this trait can aid in the improvement of varieties with better oil properties.
4. Reliability of Sources: The text references genetic studies and well-known genes (fad2A and fad2B), confirming the reliability of information sources.
Weaknesses:
1. Technical Language: The text includes specialized scientific language, making it challenging for readers without a background in plant genetics.
2. Terminology Explanation: Some abbreviations, such as RIL (Recombinant Inbred Line) or ICIM-ADD, are not explained, which could be problematic for readers outside the field of plant genetics.
3. More Context for Non-Specialist Readers: An introduction or summary explaining the significance of high oleic acid content in peanuts for society or the food industry could enhance understanding of the research context.
4. Division into Parts: The text could be divided into more understandable sections, such as introduction, methods, results, and discussion, to facilitate reading and comprehension.
Summary: The article is reliable and provides important information about the genetic basis of high oleic acid content in peanuts. However, to become more accessible to a wider audience, it would require some improvements in language and structure. For specialists in plant genetics, the article is a valuable source of information.
Comments on the Quality of English LanguageMinor editing of English language required.
Reviewer 3 Report
Comments and Suggestions for Authors
The manuscript presents research in QTL identification to improve the quality of peanut seed oil. Overall, the paper is well-written, but I have a few comments that could easily be corrected in the paper by the Authors.
1) In the Abstract, the Authors state that "Seven fatty acids were detected in both parents." I presume that the authors meant that they considered the seven fatty acids they determined in both parents in their study. Am I correct in my guess?
2) Page 3. Subsection Sample preparation and fatty acid analysis “Fatty acids were analyzed by gas chromatography (GC) as described previously with minor modifications” - please provide a literature citation.
3) Page 3. Subsection QTL mapping analysis “A genetic map was constructed in a previous study” - please provide a literature citation.
4) Page 3. Subsection Statistical analysis - for which experimental setup was the statistical analysis performed?
5) Figure 1. In the description, please complete the individual figures, e.g. C) C16:0.
6) Acronyms used in tables should be explained below the tables.
7) Table 1. Mean - of how many values?
8) Standardize notation in tables, e.g. Table 4 PVE%, Table 5 PVE(%).
9) I suggest separating the Conclusion chapter.
